**Data Availability Statement:** All relevant data are within the paper and its Supporting Information files.

**Funding:** The authors received no specific funding for this work.

# Macular pigment changes after cataract surgery with yellow-tinted intraocular lens implantation

Akira Obana[1,2]ᵒ*, Yuko Gohto[1]ᵒ, Ryo Asaoka[1]ᵒ

1 Department of Ophthalmology, Seirei Hamamatsu General Hospital, Hamamatsu, Shizuoka, Japan,
2 Photochemical Medicine Department, Photon Medical Research Center, Hamamatsu University School of Medicine, Hamamatsu, Shizuoka, Japan

ᵒ These authors contributed equally to this work.
* obana@sis.seirei.or.jp

## Abstract

### Purpose

We previously reported that macular pigment optical density (MPOD) levels decreased during a long follow-up period after clear intraocular lens (IOL) implant surgery presumably due to excessive light exposure. We examined changes in MPOD levels in the eyes that received yellow-tinted IOL implant surgery.

### Subjects and methods

This was a prospective, observational study. Fifty-five eyes of 35 patients were studied. MPOD levels were measured with a dual-wavelength autofluorescence technique on day 4; months 1, 3, and 6; and years 1 and 2 postoperatively. The average optical densities at 0˚-2˚ eccentricities (local MPODs) and total volumes of MPOD (MPOVs) in the area within 1.5˚ and 9˚ eccentricities were analyzed.

### Results

The mean local MPOD at baseline (on day 4) was 0.79 at 0˚, 0.71 at 0.5˚, 0.68 at 0.9˚, and 0.32 at 2˚. The mean MPOV within 1.5˚ and 9˚ at baseline was 2950 and 18,897, respectively. Local MPOD at 0.9˚ and 2˚ and MPOVs were slightly decreased at month 1 and increased after that. The increase reached statistical significance in local MPOD at 0.5˚ and 2˚ and MPOVs (Tukey–Kramer test). The changes in MPOV within 9˚ at year 2 [(MPOV on year 2 − MPOV on day 4) / MPOV on day 4] were from −0.21 to 1.18 (mean and standard deviation: 1.14 ± 0.28). The MPOV of 15 eyes increased more than 10% from the initial value, was maintained within 10% in 21 eyes, and deteriorated more than 10% in only 3 eyes.

### Conclusions

Local MPOD and MPOV tended to slightly decrease month 1 postoperatively and gradually increased after that, but the rates of increases in MPOD levels were small. Yellow-tinted

**Competing interests:** The authors have declared that no competing interests exist.

IOLs that have a lower transmittance of blue light might be preferable for preserving MPOD levels after surgery.

## Introduction

Visible light (wavelength: approximately 380 to 760 nm) has the potential of inducing retinal damage [1, 2]. Especially blue light, with wavelengths ranging from 380 to 500 nm, has a high potential to induce retinal damage because of its higher energy per photon than longer wave-length lights. In the retina, blue light is absorbed by lipofuscin in the retinal pigment epithelial (RPE) layer, rhodopsin in the photoreceptor layer, and other chromophores [3–5] and singlet oxygen and other radical species that damage the RPE and photoreceptors are produced [6]. These photochemical or photooxidative light–tissue reactions occur with low and sustained irradiation under the threshold value that induces thermal changes [1, 3, 7]. This situation is called blue light hazard. Acute damage to the retina by accidentally gazing at sun light, arc welding light, laser light from medical and industrial equipment, and light emission diodes used in toys has been reported [8–22]. However, the harmfulness of chronic exposure to blue light of certain wavelengths is still controversial [23–26]. Accumulative photooxidative stress by blue light theoretically induces chronic changes in the retina, and photooxidative stress is presumed to be one of the origins of retinal disorders, such as age-related macular degeneration (AMD) [27].

The human crystalline lens becomes yellowish to brown with aging and potentially blocks harmful blue light [3, 28–30] (Fig 1A). Removal of the crystalline lens by cataract surgery enhances blue light exposure to the retina (Fig 1B), and this higher level of exposure may have the potential to progress AMD via increased photooxidative stress [27, 31]. The effects of cataract surgery, however, on AMD development has not been determined because of insufficient data [32]. On the other hand, the retina has macular pigment (MP) consisting of three carotenoids, that is, lutein [(3R,3′R,6′R)-lutein], zeaxanthin [(3R,3′R)-zeaxanthin], and *meso*-zeaxanthin [(3R,3′S; *meso*)-zeaxanthin] [33, 34]. The maximum absorption wavelength of MP is 460 nm [35] and MP acts as a filter by absorbing blue light (Fig 1B). It potentially protects against light-induced oxidative damage in the retina by quenching oxygen radicals [36–38]. Thus, the

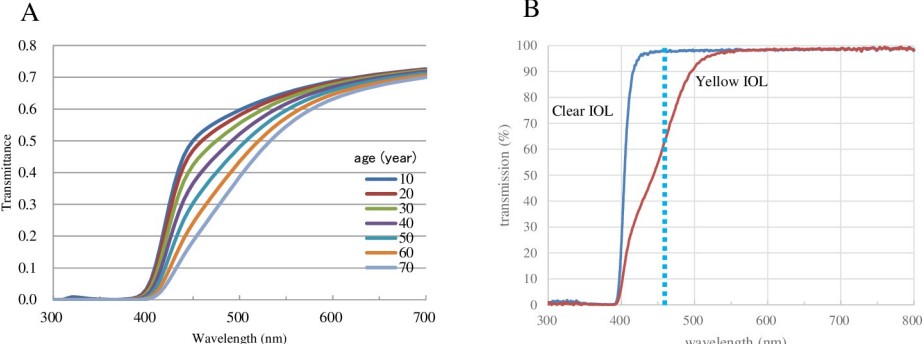

**Fig 1. A.** Transmission of human crystalline lens with age ranged 10 to 70 years old. The data is a personal gift from Okuno T. Transmission shows a monotonous decrease in blue range with age. **B.** Transmission curves of +20 diopter clear intraocular lens (IOL, SA60AT, Alcon Inc.) and yellow-tinted intraocular lenses (SN60AT). Yellow-tinted IOL blocks blue light. Dotted line indicates the maximum absorption wavelength of macular pigment (460 nm). The data is a personal gift from Tanito M.

MP is an important internal defense system in the retina. We previously investigated the changes in MP in eyes that received cataract surgery with intraocular lens (IOL) implantation. The results showed that MP levels in the eyes implanted with a clear IOL deteriorated 1 and 2 years after surgery, whereas MP levels in the eyes with a yellow-tinted IOL implantation had no change. The mechanisms underlying the deterioration of the MP in eyes with a clear IOL are unknown, but we speculated that a high transmission rate of short wavelength components with a clear IOL might induce higher exposure of blue light to the MP and cause high consumption of MP carotenoids [39] (Fig 1B). We considered that a yellow-tinted IOL has advantages in preserving the MP. While in contrast, Nolan et al. reported increased MP optical density (MPOD) levels in the eyes with yellow-tinted IOL implants and unchanged MPOD levels in the eyes with clear IOLs [40]. Therefore, the changes in MP in the eyes that received IOL implants are still controversial. In our previous study, we used resonance Raman spectroscopy (RRS) to measure MP levels, and Nolan et al. used heterochromatic flicker photometry. RRS is a highly specific method that can measure MPOD levels because this method directly measures the Raman signal from the MP that is specific to carotenoids. However, RRS allows counting the total Raman signals from the MP within a diameter of 1 mm of the central fovea. Actually, the MP was distributed more widely in the macular region, and the importance of measuring the total amount of MP at the macular region is proposed to investigate the protective effects of MP [41]. Heterochromatic flicker photometry has been widely used for the investigation of MP for a long time, but its subjective nature needs subjective cooperation and may induce some bias. Heterochromatic flicker photometry also measures the MP at limited lesions of the retina. In contrast, a dual-wavelength autofluorescent technique is an objective technique. With this technique, not only MPOD at any location of the macular area but also a total volume of MPOD in the area within any eccentricity can be obtained. A module on the Heidelberg SPECTRALIS with MultiColor (Spectralis-MP, Heidelberg Engineering Inc.) equipped with a dual-wavelength (486 and 518 nm) autofluorescent technique was developed recently, and the reliability of this device was validated [41–43].

In this study, we measured MPOD levels in eyes implanted with yellow-tinted IOLs 4 days to 2 years after surgery using the dual-wavelength autofluorescence technique to verify our previous results.

## Subjects and methods

Fifty-five eyes of 35 patients (18 men and 17 women) who underwent cataract surgery at Seirei Hamamatsu General Hospital between September 2016 and September 2018 were included in the study (Table 1). Patient ages ranged from 41 to 86 years, and the mean and standard

Table 1. Demographics of the patients.

| | Male | Female |
|---|---|---|
| patients | 18 | 17 |
| eyes | 27 | 28 |
| Age range (years) | 56–86 | 41–85 |
| Mean age (SD) (years) | 69.0 (8.4) | 71.1 (10.3) |
| No. of eyes implanted the following IOLs | | |
| SN60WF | 14 | 19 |
| XY1 | 9 | 7 |
| YP2.2 | 2 | 2 |
| PCB00V | 2 | 0 |

SD, standard deviation; IOL, intraocular lens

deviation was 71.2 ± 9.6 years. All surgeries were performed by three experienced surgeons (AO, YG, and HS) using the same phaco instrument (Infinity, Alcon Inc.) and operating microscope (OPMI Visu210/S88, Carl Zeiss, Oberkochen, Germany). The operating technique was the same reported in the previous study. IOLs were implanted in the capsuler bag in all eyes. The products of IOLs that were implanted were SN60WF (Alcon Inc., Fig 1B) in 33 eyes, XY1 (HOYA Corporation) in 16 eyes, YP2.2 (Kowa Company, LtD) in 4 eyes, and PCB00V (AMO Japan K.K) in 2 eyes. All eyes were free of disorders that could have affected the MPOD measurements, including corneal and vitreal opacities, and retinal disorders such as vein occlusion, multiple drusen, and pigment abnormalities in the macula. There was no severe inflammation postoperatively.

MPOD levels were measured with a SPECTRALIS OCT with MultiColor equipped with a dual-wavelength autofluorescence technique. The actual measurement was performed in the same manner as described in previous studies [44–46]. The reference plateau for assumed absence of MP was set at 9˚ eccentricity. The measurements were performed in eyes under mydriasis induced by 2.5% phenylephrine hydrochloride and 1% tropicamide. The average optical densities at 0˚, 0.5˚, 0.9˚, and 2˚ eccentricities (local MPODs) and total MPOD volumes (MPOVs) in the area within 1.5˚ and 9˚ eccentricities were analyzed. The RRS technique obtained total Raman counts in the central area with a 1-mm diameter. In order to adjust the RRS measurement, MPOVs in the area of 1.5˚ were analyzed. Patients were examined on day 4; months 1, 3, and 6; and years 1 and 2, postoperatively. Patients underwent usual ophthalmological examinations including visual acuity testing and measurement of intraocular pressure at every time point. Posterior capsule opacification was evaluated with slit-lamp biomicroscopy, and no eyes had clinically relevant opacification that could have caused loss of visual acuity and light transmission for MPOD measurements. It was confirmed verbally that no patients started to take supplements containing lutein and/or zeaxanthin after surgery.

These prospective case series were approved by the institutional review board of Seirei Hamamatsu General Hospital (IRB No. 2199). The protocol followed the tenets of the Declaration of Helsinki. All patients provided written informed consent at enrollment.

## Statistical analysis

The values of local MPODs at 0˚, 0.5˚, 0.9˚, and 2˚ eccentricities and MPOVs in the areas within 1.5˚ and 9˚ eccentricities were compared at day 4; months 1, 3, and 6; and years 1 and 2, using a linear mixed model with patients as a random effect (because one or two eyes of a patient were included) and the Tukey–Kramer test, which considered multiple comparisons. The linear mixed model is equivalent to an ordinary linear regression in that the model describes the relationship between the predictor variables and a single outcome variable. However, standard linear regression analysis rests on the assumption that all observations are independent of each other. In the current study, measurements were nested within subjects and hence were dependent on each other. Ignoring this grouping of the measurements will result in underestimation of the standard errors of regression coefficients. The linear mixed model adjusts for the hierarchical structure of the data, modeling in such a way that measurements are grouped within subjects to reduce the possible bias derived from the nested structure of the data [47, 48]. Subsequently, these analyses were iterated using only eyes that were subjected to measurements both at day 4 and year 1 and day 4 and year 2.

In addition, the change rate of MPOV within 9˚ on year 2 was calculated as (MPOV on year 2 – MPOV on day 4) / MPOV on day 4. The change rates were compared between SN60WF and XY1 using a non-paired t-test.

All statistical analyses were performed using the statistical programming language R (ver. 3.6.1, The R Foundation for Statistical Computing, Vienna, Austria); StatView, version 5.0; SAS statistical software (Cary, NC); and Statistical Package for Service Solution (SPSS) software, version 26 (IBM SPSS, Chicago, IL); $p < 0.05$ was considered significant.

## Results

The MPOD levels were measured in all eyes on day 4 postoperatively, which was defined as the baseline value of MPOD levels in each eye. Some eyes failed to receive MPOD measurements at some time points, and the number of eyes that received MPOD measurements was 49 at month 1, 51 at month 3, 47 at month 6, 48 at year 1, and 39 at year 2. Local MPOD levels at four eccentricities and MPOV within two eccentricities for all time points are shown in Table 2. The differences in local MPOD levels and MPOV between day 4 and year 1, postoperatively, were analyzed statistically in 48 eyes of 29 subjects who received MPOD measurements at year 1 (Table 3). Local MPOD levels increased significantly at year 1 at 0˚, 0.5˚, and 2˚ eccentricities and MPOV within 1.5˚ and 9˚ eccentricities increased significantly at year 1. The differences in local MPOD levels and MPOV between day 4 and year 2 postoperatively were also statistically analyzed in 39 eyes of 23 subjects who received MPOD measurements at year 2 (Table 4). A significant increase was observed in local MPOD at 0.5˚ and 2˚ and MPOV within 1.5˚ and 9˚ eccentricities.

MPOD levels were measured at all time points during a postoperative follow-up period of 2 years in 35 eyes of 22 subjects. Fig 2 shows changes in the mean local MPOD levels in these eyes. There were no significant changes in local MPOD at 0˚ and 0.5˚ between any two time points. The local MPOD at 0.9˚ at month 1 was lower than that at day 4, month 6, and years 1 and 2. Local MPOD at 2˚ at month 1 was lower than that at day 4, month 6, and years 1 and 2, and the differences between month 1 and month 6, year 1, and year 2 were significant. Local MPOD at 2˚ at year 1 was significantly higher than that at day 4 (Tukey–Kramer test). Figs 3 and 4 show changes in MPOV within 1.5˚ and 9˚, respectively. MPOV within 1.5˚ at year 2

**Table 2. Local MPOD levels and MPOV at four eccentricities at all time points.**

|  |  | Day 4 | Month 1 | Month 3 | Month 6 | Year 1 | Year 2 |
|---|---|---|---|---|---|---|---|
| 0˚ | Range | 0.37–1.39 | 0.31–1.38 | 0.44–1.30 | 0.44–1.48 | 0.39–1.26 | 0.45–1.33 |
|  | Mean (SD) | 0.79 (0.21) | 0.82 (0.24) | 0.82 (0.21) | 0.82 (0.23) | 0.82 (0.21) | 0.80 (0.22) |
| 0.5˚ | Range | 0.21–1.12 | 0.25–1.24 | 0.21–1.21 | 0.27–1.24 | 0.24–1.27 | 0.32–1.24 |
|  | Mean (SD) | 0.71 (0.22) | 0.72 (0.22) | 0.72 (0.22) | 0.72 (0.21) | 0.72 (0.22) | 0.72 (0.21) |
| 0.9˚ | Range | 0.21–1.12 | 0.22–1.05 | 0.21–1.04 | 0.25–1.05 | 0.24–1.05 | 0.34–1.04 |
|  | Mean (SD) | 0.68 (0.22) | 0.66 (0.21) | 0.67 (0.21) | 0.68 (0.21) | 0.69 (0.21) | 0.67 (0.20) |
| 2˚ | Range | 0.06–0.74 | 0.08–0.68 | 0.06–0.71 | 0.09–0.71 | 0.08–0.72 | 0.12–0.68 |
|  | Mean (SD) | 0.32 (0.17) | 0.33 (0.17) | 0.33 (0.17) | 0.33 (0.17) | 0.34 (0.17) | 0.31 (0.14) |
| MPOV within 1.5˚ | Range | 846–5352 | 977–5073 | 846–5051 | 1084–4992 | 1013–5014 | 1397–4950 |
|  | Mean (SD) | 2950 (1003) | 2960 (1010) | 2988 (981) | 2996 (981) | 3042 (1005) | 2951 (917) |
| MPOV within 9˚ | Range | 4948–42810 | 4842–71142 | 4948–41430 | 5886–42042 | 7026–42980 | 7118–37789 |
|  | Mean (SD) | 18,897 (9294) | 19,800 (11,724) | 19,168 (9300) | 19,447 (9340) | 19,707 (9374) | 17,886 (7628) |

SD, standard deviation

**Table 3. Comparison of MPOD levels between day 4 and year 1 postoperatively.**

| | Day 4 | | | Year 1 | | | |
|---|---|---|---|---|---|---|---|
| | **Mean** | **Standard deviation** | **Range** | **Mean** | **Standard deviation** | **Range** | **P value** |
| Local MPOD | | | | | | | |
| 0˚ | **0.77** | 0.21 | 0.37–1.39 | **0.82** | 0.21 | 0.39–1.26 | **0.0104** |
| 0.5˚ | **0.70** | 0.22 | 0.21–1.19 | **0.72** | 0.22 | 0.24–1.27 | **0.0252** |
| 0.9˚ | 0.67 | 0.22 | 0.21–1.12 | 0.69 | 0.21 | 0.24–1.05 | 0.0924 |
| 2˚ | **0.32** | 0.17 | 0.06–0.74 | **0.34** | 0.17 | 0.08–0.72 | **0.0029** |
| MPOV within 1.5˚ | **2936** | 1012 | 846–5352 | **3042** | 1005 | 1014–5014 | **0.0045** |
| MPOV within 9˚ | **18,767** | 9591 | 4948–42,810 | **19,707** | 9374 | 7026–42,980 | **0.0036** |

Significant values were shown in bold.

was significantly higher than that at day 4 and months 1 and 3. MPOV within 1.5˚ at month 1 was significantly lower than that at month 3, year 1 and 2. MPOV within 9˚ at year 1 was significantly higher than that at day 4. MPOV within 9˚ at month 1 was significantly lower than that at month 6, and year 1 and 2. (Tukey–Kramer test).

The change rate of MPOV within 9˚ at year 2 was from −0.21 to 1.18. The mean change rate and standard deviation was 1.14 ± 0.28. We divided change rates into three categories: +1.1 and more, lower than +1.1 and higher than −1.1, and −1.1 and lower. Fifteen eyes had an increased MPOV of >10% from the initial value. The change rates of 21 eyes were within 10%, and only three eyes had a deteriorated MPOV of more than 10%. The mean change rates of each IOL product were 0.09 ± 0.02 with SN60WF, 0.23 ± 0.18 with XY1, 0.08 with YP2.2, and −0.04 with PCB00V. There was no significant difference in the mean change rates between SN60WF and XY1 (p = 0.478, non-paired *t*-test). Statistical analyses were not performed in YP2.2 and PCB00V because of the small number of eyes that were implanted.

## Discussion

The present results showed that MPOD levels in the yellow-tinted IOL-implanted eyes tended to be slightly decreased at month 1 postoperatively and gradually increased after that. The increase in local MPOD reached statistical significance at 0˚, 0.5˚, and 2˚ at year 1 (Table 3) and at 0.5˚ and 2˚ at year 2 (Table 4). MPOV within 1.5˚ and 9˚ also increased significantly at years 1 and 2 (Tables 3 and 4). Twenty-five eyes (64% of the eyes measured MPOD at year 2) showed higher MPOVs within 9˚ at year 2 than those at day 4, and among them, 15 eyes (38%) increased the MPOV >10%, whereas only 3 eyes had a deteriorated MPOV of more than 10%.

**Table 4. Comparison of MPOD levels between day 4 and year 2 postoperatively.**

| | Day 4 | | | Year 2 | | | |
|---|---|---|---|---|---|---|---|
| | **Mean** | **Standard deviation** | **Range** | **Mean** | **Standard deviation** | **Range** | **P value** |
| Local MPOD | | | | | | | |
| 0˚ | 0.77 | 0.21 | 0.37–1.39 | 0.80 | 0.22 | 0.45–1.33 | 0.313 |
| 0.5˚ | **0.69** | 0.22 | 0.21–1.19 | **0.72** | 0.21 | 0.32–1.24 | **0.0204** |
| 0.9˚ | 0.65 | 0.21 | 0.21–1.12 | 0.67 | 0.20 | 0.34–1.04 | 0.135 |
| 2˚ | **0.29** | 0.15 | 0.06–0.74 | **0.31** | 0.14 | 0.12–0.68 | **0.0188** |
| MPOV within 1.5˚ | **2817** | 973 | 846–5352 | **2951** | 917 | 1392–4950 | **0.0086** |
| MPOV within 9˚ | **16,991** | 8504 | 4948–40,224 | **17,886** | 7628 | 7118–37,789 | **0.0408** |

Significant values were shown in bold.

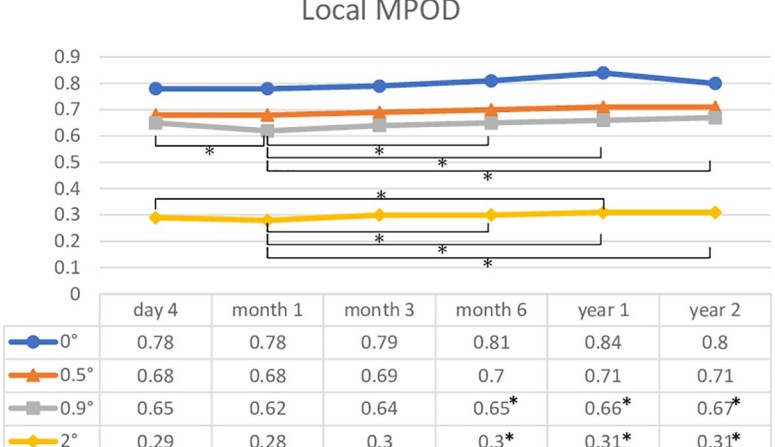

**Fig 2. A change in the mean local macular pigment optical density (MPOD) levels in the eyes that had all measurement data during a postoperative follow-up period of 2 years (35 eyes of 22 subjects).** Asterisks indicate a significant difference ($p < 0.05$) in local MPOD levels between two time points.

Nolan et al. measured MPOD at four eccentricities, i.e., 0.25˚, 0.5˚, 1˚, and 1.75˚, with flicker photometry at 12 months after surgery in 11 eyes with yellow-tinted IOLs. They confirmed the increase in local MPOD levels at 0.25˚ and 0.5˚. The increase in MPOD levels at 1˚ and 1.75˚ did not reach a significant level, but the mean MPOD of these four eccentricities increased significantly at months 3, 6, and 12. They speculated that the mechanisms for the increased MPOD were as follows: yellow-tinted IOLs have a lower transmission in the blue light region than clear IOLs, but the transmission was still higher than that in the human lenses of elderly people (Fig 1), and the increased visible light irradiation of the retina after cataract surgery could stimulate enhanced retinal capture of circulating lutein and zeaxanthin, perhaps due to a mechanism involving increased isomerization of the MP constituents under light exposure [7]. Because the living human body generally has an autoregulatory system, it seems reasonable that the increase in MPOD can prevent retinal damage caused by increased short wavelength visible light intensities and correspondingly higher oxidative stress.

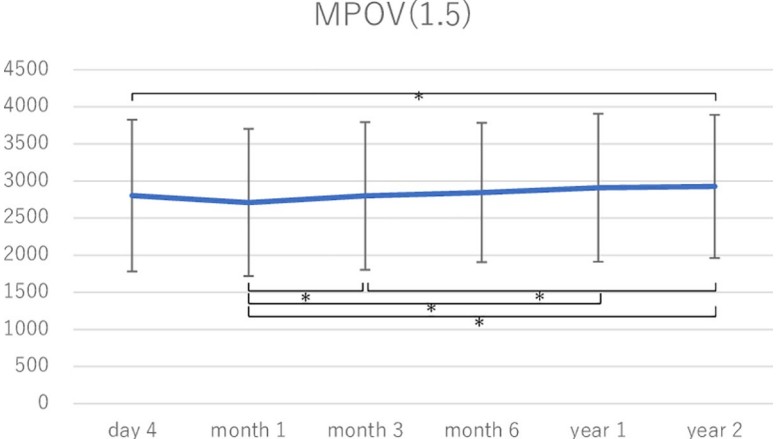

**Fig 3. A change in macular pigment optical density volume (MPOV) within 1.5˚ eccentricity in the eyes that had all measurement data during a postoperative follow-up period of 2 years (35 eyes of 22 subjects).** Asterisks indicate a significant difference ($p < 0.05$) in MPOV between two time points.

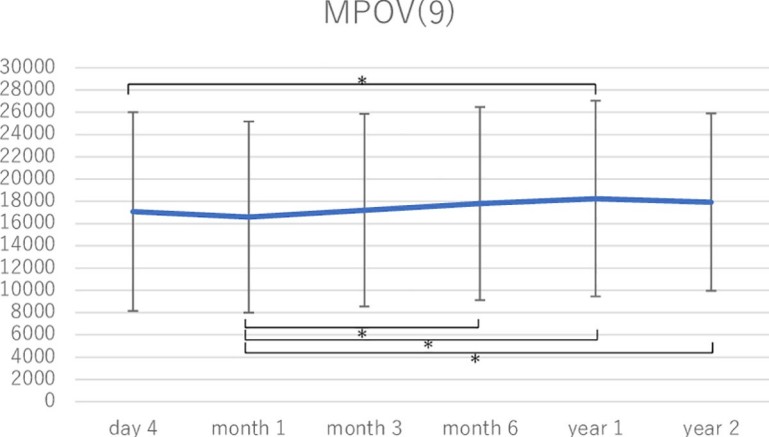

**Fig 4. A change in macular pigment optical density volume (MPOV) within 9˚ eccentricity in the eyes that had all measurement data during a postoperative follow-up period of 2 years (35 eyes of 22 subjects).** Asterisks indicate a significant difference (p < 0.05) in MPOV between two time points.

The present results also showed increases in MPOD levels after surgery at some eccentricities and MPOV, but the increased rates were quite different from those reported in a study by Nolan et al. In the study by Nolan et al., the increased value of MPOD was 0.16 at 0.25˚ and 0.123 at 0.5˚. The initial MPOD levels at 0.25˚ on week 1 ranged from 0.07 to 0.52 (mean ± SD, 0.28 ± 0.17), and then the increased rate in the mean MPOD at 0.25˚ was 57%. In the present study, the increased rates at year 1 were 6% at 0˚ and 3% at 0.5˚. One possible reason for the difference in change rates is the different levels of the initial MPOD. Our initial MPOD levels were high (0.77 at 0˚ and 0.7 at 0.5˚) compared with those of Nolan's study (0.28 at 0.25˚). Optical density of 0.7 represents a transmission rate of 17%, whereas that of 0.28 represents a transmission rate of 52%. In a study by Nolan et al., the mean MPOD at 0.25˚ at year 1 became 0.44, which represented a transmission rate of 36%. Our high initial MPOD levels might be sufficient to protect against blue light, and the increase in MP might not be necessary as an autoregulatory system. However, the measurement techniques used to calculate the initial MP values differed in the two studies, therefore, we should take into consideration of the difference in MP values, although the differences were thought to be minor. The increased rates should be studied further in large number of subjects to draw a confirm conclusion. Nolan et al. did not measure MPOD at month 1. Therefore, the transient decrease in MPOD around month 1 could not be compared.

In our previous study using RRS to measure MPOD, MPOD levels remained stable in the eyes with yellow-tinted IOL implants for 2 years. MPOV within 1.5˚ eccentricity in the present study was comparable with the measurement value by RRS, and it was significantly increased, although the increase rates were small (3.6% at year 1 and 4.7% at year 2). The reason for this discrepancy is unclear. One possible reason was the influence of the cataract that induced underestimation of MPOD in RRS measurement. The RRS measurement is directly influenced by the media opacity because RRS counts Raman signal intensities of macular carotenoids produced by the irradiation of a blue exciting light. With the autofluorescence technique, the influence of the media opacity can be canceled because MPOD is indirectly obtained via the comparison of lipofuscin fluorescence intensities of peripheral and central areas that are influenced equally by the media opacity. Although no eyes had clinically relevant opacification by slit-lamp biomicroscopy in our previous study, the influence of mild opacification could not be denied. Another possibility is the small number of subjects. Two-thirds of the subjects

showed increased MPOD, but one-third showed decreased MPOD in the present study. Therefore, the change rate may vary in a larger sample size.

The limitation of this study was a small number of subjects, as described earlier, and there was no comparison of eyes implanted with a clear IOL. Further study is needed to confirm the increased rate of MPOD levels in the eyes with yellow-tinted IOL implants. However, in the eyes with yellow-tinted IOL implants, there seemed to be no deterioration in MPOD as observed in the eyes with clear IOL implants.

## Supporting information

**S1 Data.**
(XLSX)

## Acknowledgments

The authors would like to thank Tutomu Okuno of National Institute of Occupational Safety and Health, Japan (retired) and Masaki Tanito of Shimane University Faculty of Medicine for giving the data on spectral transmission of human lens and intraocular lens.

## Author Contributions

**Conceptualization:** Akira Obana.

**Data curation:** Akira Obana.

**Formal analysis:** Akira Obana, Ryo Asaoka.

**Investigation:** Akira Obana, Yuko Gohto.

**Methodology:** Akira Obana, Yuko Gohto.

**Project administration:** Akira Obana, Yuko Gohto.

**Resources:** Akira Obana.

**Software:** Akira Obana.

**Writing – original draft:** Akira Obana.

**Writing – review & editing:** Ryo Asaoka.

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
