## [Decision Letter · Decision Letter 0]

11 Feb 2021

PONE-D-20-39598

Macular pigment changes after cataract surgery with yellow-tinted intraocular lens implantation

PLOS ONE

Dear Dr. Obana,

Thank you for submitting your manuscript to PLOS ONE. After careful consideration, we feel that it has merit but does not fully meet PLOS ONE’s publication criteria as it currently stands. Therefore, we invite you to submit a revised version of the manuscript that addresses the points raised during the review process.

Please add a schematic of the absorbance spectra the ocular media (cornea, lens, retina [e.g. MP and other chromophores]) to help readers understand the region of high-energy, short-wavelength light. Please provide spectral transmission of the SN60 WF, XY1, YP2.2 and PCB00V IOLs.

We look forward to receiving your revised manuscript.

Kind regards,

Alfred S Lewin, Ph.D.

Academic Editor

PLOS ONE

Journal Requirements:

2. Please provide a table of patient demographics.

Reviewers' comments:

Reviewer's Responses to Questions

**Comments to the Author**

1. Is the manuscript technically sound, and do the data support the conclusions?

Reviewer #1: Yes

2. Has the statistical analysis been performed appropriately and rigorously? 

Reviewer #1: Yes

3. Have the authors made all data underlying the findings in their manuscript fully available?

Reviewer #1: Yes

4. Is the manuscript presented in an intelligible fashion and written in standard English?

Reviewer #1: Yes

5. Review Comments to the Author

Reviewer #1: 1) Adding a schematic of the absorbance spectra the ocular media would be beneficial (cornea, lens, retina [e.g. MP and other chromophores]) to help readers understand the region of high-energy, short-wavelength light

2)The authors have correctly pointed it out but direct comparisons of RRS and 2-wavelength AF and HFP are inappropriate. Differences in measured MPOD among the 3 are difficult to assign meaning to.

3) Please provide spectral transmission of the SN60 WF, XY1, YP2.2 and PCB00V IOLs. This might help the reader understand the absorption contribution relative to the other ocular components

4) Although the author states that "No L/Z supplementation was utilized, a dietary questionnaire control might be useful

5) Tbl 2 and 3 show that virtually all of the MPOD and MPOV values increased at every position. As MP cannot be synthesized de novo and must be obtained through diet, I'm interested to learn more about the etiology. I ask out of clinical interest because the theory posited in Lines 74-79 states that increased transmission of SW light leads to decreased MP but the theory described in Lines 233-239 state that increased retinal irradiation following cataract surgery leads to enhanced retinal capture of circulating L/Z due to increased isomerization. Should all ECCE patients be using L/Z supplementation?

6) Lines 127-129 discussed analysis of RRS and MPOV @ 1.5 degrees measured through FAF. Comparison of these values may be inappropriate due to technique differences.

7) A table displaying change rate for time pints and eccentricities would be valuable

6. PLOS authors have the option to publish the peer review history of their article (what does this mean?). If published, this will include your full peer review and any attached files.

Reviewer #1: No

---

## [Author Response · Author response to Decision Letter 0]

25 Feb 2021

Thank you for organizing a quick review. We appreciate the reviewers’ comments and have revised our manuscript accordingly. 

The following details are our point-by-point responses (black font) to the reviewers’ comments (blue font, Italic). The main revised parts are indicated in the proofreading file in red. 

We checked our manuscript according to PLOS ONE’s style and we believe this manuscript fits it. If not, please let us know. We will revise it quickly.

2. Please provide a table of patient demographics.

Thank you for your suggestion. We added patient’s demographic data in Table 1.

3. We note that you have indicated that data from this study are available upon request. PLOS only allows data to be available upon request if there are legal or ethical restrictions on sharing data publicly.

We uploaded our data set.

Dear reviewer #1

We appreciate your immensely thorough review with positive comments and useful suggestions. We have analyzed our data and revised our manuscript according to your suggestions. The following details are our point-by-point responses to your comments. 

1) Adding a schematic of the absorbance spectra the ocular media would be beneficial (cornea, lens, retina [e.g. MP and other chromophores]) to help readers understand the region of high-energy, short-wavelength light

Thank you for your comment. We added figures of spectral transmission of human crystalline lenses of different age and clear and yellow-tinted IOLs. The maximum absorption wavelength of macular pigments is 460 nm and this information was added in Figure 1B.

2)The authors have correctly pointed it out but direct comparisons of RRS and 2-wavelength AF and HFP are inappropriate. Differences in measured MPOD among the 3 are difficult to assign meaning to.

We consider that this comment is in response to the discussion from line 243 to 256. 

Both two wavelength AF and HFP measure absorbance of MP, i.e. both methods measure the same thing. Therefore, the values measured by two methods should be identical, but actually there is some difference in the absolute values, although the values by two methods show high correlation. In this discussion, we compared not the absolute values but the increase rates between our study and Nolan’s study and we think this comparison was reasonable. 

However, we attributed the different increase rates in two studies to the difference in the initial MP absolute values. This discussion might have some bias. Therefore, we added the following sentence in this discussion, “However, the measurement techniques used to calculate the initial MP values differed in the two studies, therefore, we should take into consideration of the difference in MP values, although the differences were thought to be minor.” 

3) Please provide spectral transmission of the SN60 WF, XY1, YP2.2 and PCB00V IOLs. This might help the reader understand the absorption contribution relative to the other ocular components

Thank you for your suggestion. We added the spectral transmission of clear and yellow-tinted IOLs in Figure 1B.

4) Although the author states that "No L/Z supplementation was utilized, a dietary questionnaire control might be useful

Thank you for your comment. We asked the patients at the time of every examination and confirmed that they didn’t start taking supplements including lutein and zeaxanthin. We added the following sentence, “It was confirmed verbally that no patients started to take supplements containing lutein and/or zeaxanthin after surgery.”

5) Tbl 2 and 3 show that virtually all of the MPOD and MPOV values increased at every position. As MP cannot be synthesized de novo and must be obtained through diet, I'm interested to learn more about the etiology. I ask out of clinical interest because the theory posited in Lines 74-79 states that increased transmission of SW light leads to decreased MP but the theory described in Lines 233-239 state that increased retinal irradiation following cataract surgery leads to enhanced retinal capture of circulating L/Z due to increased isomerization. Should all ECCE patients be using L/Z supplementation?

Thank you for your comment. The mechanisms of increase in MP after surgery is unknown. But considering the autoregulatory system of humans, this result seems reasonable. Nolan et al speculated that “the increased visible light irradiation of the retina after cataract surgery could stimulate enhanced retinal capture of circulating lutein and zeaxanthin, perhaps due to a mechanism involving increased isomerization of the MP constituents under light exposure” as described in Discussion. We have no evidences to support their speculation. We think that whether MP increase or decrease depends on the extent of increased irradiation. When the irradiation is excessive (for example, in case with clear IOL implantation), consumption of MP may be higher than MP accumulation, and MP decreases. 

6) Lines 127-129 discussed analysis of RRS and MPOV @ 1.5 degrees measured through FAF. Comparison of these values may be inappropriate due to technique differences.

RRS uses a different principle from AF. It measures the Raman signal of carotenoids in MP in the area of a diameter of 1 mm (= 1.6̊ ). Therefore, we compared the increase rate of MP in the similar area, i.e. MPOV within 1.5̊ eccentricity in AF and total Raman signals within 1.6̊ eccentricity. Since we compared not the absolute values but the rate of values, we consider this comparison is reasonable.

---

## [Editor Report · Decision Letter 1]

1 Mar 2021

Macular pigment changes after cataract surgery with yellow-tinted intraocular lens implantation

PONE-D-20-39598R1

Dear Dr. Obana,

We’re pleased to inform you that your manuscript has been judged scientifically suitable for publication and will be formally accepted for publication once it meets all outstanding technical requirements.

Kind regards,

Alfred S Lewin, Ph.D.

Section Editor

PLOS ONE
---

## [Editor Report · Acceptance letter]

8 Mar 2021

PONE-D-20-39598R1 

Macular pigment changes after cataract surgery with yellow-tinted intraocular lens implantation. 

Dear Dr. Obana:

I'm pleased to inform you that your manuscript has been deemed suitable for publication in PLOS ONE. Congratulations! Your manuscript is now with our production department. 

Kind regards, 

on behalf of

Dr. Alfred S Lewin 

Section Editor

PLOS ONE